# Sinking of microbial-associated microplastics in natural waters

Thu Ha Nguyen *, Fiona H. M. Tang, Federico Maggi

Laboratory for Advanced Environmental Engineering Research, School of Civil Engineering, The University of Sydney, Sydney, New South Wales, Australia

* thuha.nguyen@sydney.edu.au

## Abstract

Degraded plastic debris has been found in nearly all waters within and nearby urban developments as well as in the open oceans. Natural removal of suspended microplastics (MPs) by deposition is often limited by their excess buoyancy relative to water, but this can change with the attachment of biological matter. The extent to which the attached biological ballast affects MP dynamics is still not well characterised. Here, we experimentally demonstrate using a novel OMCEC (Optical Measurement of CEll colonisation) system that the biological fraction of MP aggregates has substantial control over their size, shape and, most importantly, their settling velocity. Polyurethane MP aggregates made of 80% biological ballast had an average size almost twice of those containing 5% biological ballast, and sank about two times slower. Based on our experiments, we introduce a settling velocity equation that accounts for different biological content as well as the irregular fractal structure of MP aggregates. This equation can capture the settling velocity of both virgin MPs and microbial-associated MP aggregates in our experiment with 7% error and can be used as a preliminary tool to estimate the vertical transport of MP aggregates made of different polymers and types of microbial ballast.

## Introduction

The global plastic production has reached 348 million tonnes in 2017 [1] for household, retail, and industrial demands. However, plastic production and use have not been accompanied by efficient plastic waste management, and consequently, 1 to 4% of this plastic amount reaches the ocean every year [2, 3]. Microplastics (MPs) with size smaller than 5 mm is the most common plastic debris found in marine habitats [4]. MPs consist of polymers with different densities; the most commonly produced polymers belong to polypropylene (PP) and polyethylene families (LDPE, LLDPE, MDPE, HDPE) with density ranging from 0.83 to 0.97 g cm$^{-3}$, followed by polyvinyl chloride (PVC), polyurethane (PUR), and polyethylene terephthalate (PET) with density ranging from 1.16 to 1.56 g cm$^{-3}$ [1, 5]. MPs contain toxic additives like phthalate, organotin, nonylphenol, polybrominated diphenyl ethers (PBDE), and triclosan, which can be released if degraded chemically or biologically [6, 7]. MPs have the peculiarity to sorb and be the carrier agent of anthropogenic pollutants such as dichloro-diphenyl-trichloroethane

**Funding:** THN: Australia Awards Scholarship. Funder: Australia Department of Foreign Affair and Trade. (https://dfat.gov.au). F.M and F.H.M.T:

SREI2020 Envirosphere, The University of Sydney. (https://sydney.edu.au/). F.M – SOAR, The University of Sydney (https://sydney.edu.au/). Did the sponsors or funders play any role in the study design, data collection and analysis, decision to publish, or preparation of the manuscript? NO.

**Competing interests:** The authors have declared that no competing interests exist.

(DDT), polychlorinated biphenyl (PCBs), and phenanthrene [8–10], but can also leach other embedded organic compounds known to be metabolized by attached microorganisms [11]. Having sizes similar to plankton, MPs can become part of the aquatic micro-ecosystems, favour microbial activity, be transferred to the higher trophic levels, and carry harmful chemicals into the food web as substantiated by more than 245 scientific studies [12].

Various governance strategies and regulations have been enacted over the years to reduce the sources of pollution affecting marine environments. The first global legislation is the International Convention for the Prevention of Pollution from Ships (MARPOL 73/78) signed in 1973 by 134 countries, but this alone has not effectively reduced plastic pollution in open waters. More recently, the G7 and the EU Community have committed to sustainable plastic management by increasing plastic reuse and reducing plastic waste [13, 14]. These, along with other initiatives, have received growing attention and are contributing to lowering plastic and MP pollution in some jurisdictions such as through the ban of lightweight plastic bags and microbeads in cosmetics [15–18]. At a larger scale, the UN Environment Programme has established strategies and guidelines to increase the global awareness of plastic pollution and recycling, as well as improve global plastic waste management [19–21]. More initiatives are being conceptualized or are under scrutiny to enhance their effectiveness, but we underline that they should be accompanied by research incentives to increase our understanding of MP residence time in the water column, which is crucial to quantify impacts on the ecosystem [22] and develop remediation plans.

MP pollution has received growing attention in the scientific community in the past decade, but the majority of research efforts have concentrated on MP sources and their impacts on aquatic organisms and human health, while little attention has been paid on understanding their movement in the environment and open waters in particular. MP residence time in the water column is the most crucial factor determining whether MPs are naturally removed by settling and burying within deposited sediment [23] or if they remain in suspension and escalate the bioaccumulation pathway along the food chain [24]. The residence time of a particle is conventionally estimated from its settling velocity, which depends on its physical properties and hydrodynamics. The particle properties such as size and internal geometry determine the sinking rate, but these properties can be altered by the growth of biological ballast [25], i.e., the attached microorganisms and their metabolic byproducts; hydrodynamic effects of this ballast can have substantial repercussions on the overall mass flux. Earlier studies [26–29] have quantified the sinking rate of MPs incorporating biological matter but no study up to now has simultaneously measured the sinking rate and the biological fraction on individual aggregates made of multiple materials. This leaves an important knowledge gap in the understanding of the mechanisms that control MP dynamics along the water column.

Here, we aim to investigate the extent to which biological ballast can alter the geometrical properties and terminal velocity of MP aggregates. We therefore question what impacts microbial colonisation can have on MP residence time in natural waters. We conducted this investigation using our unique fully-controlled settling column (largely described in [30, 31]) equipped with the new OMCEC (Optical Measurement of CEll colonisation) system developed and tested as described in [32]. OMCEC simultaneously performs micro Particle Tracking Velocimetry ($\mu$PTV) and light spectral analysis, and instantaneously measures three quantities of interest for individual microparticles including material composition, geometric properties (size and shape), and motion (direction and speed) in a non-invasive and non-destructive way. We tested polyurethane MPs within the size range of 90 to 300 $\mu$m incubated with microorganisms sampled from a brackish river in Australia. Using these data, we developed and tested a terminal velocity equation that considers the shape irregularity and fractal structures of microbial-associated MP aggregates.

## Materials and methods

### Material preparation

MPs were generated from blue Polyurethane (PUR) pipes (RS. 483-5765) by manual sawing; sawdust was sieved to sizes between 90 and 300 $\mu$m. Sawing emulated the genesis of irregularly shaped MPs due to physical/mechanical forcing, which is widely observed in natural waters [33].

Natural microorganisms were sampled from the top 10 cm of the Hawkesbury River, NSW, Australia, in August 2018 (winter) using a plankton net with 20 $\mu$m aperture. The sampling site (33°24.12' S, 151° 0.3' E) is about 50 km upstream of the ocean and the measured brackish water density was 1.007 g/cm$^3$. The sampled biological matter included free-living microorganisms larger than the mesh size as well as aggregates of microrganisms of different types and sizes. Based on microscope observation (S1a Fig) and the literature on the Hawkesbury River ecosystems [34], the sampled microorganisms may include freshwater/brackish diatom genus *Cyclotella*, marine planktonic diatoms *Thalassiosira* spp., *Ditylum brightwellii, Rhizosolenia setigera, Pseudo-nitzschia pungens* and *Chaetoceros* spp, and bacteria that attached on the aggregates. The natural microorganism sample was transferred into two 600 mL glass beakers with 100 mL in each beaker and was diluted with Hawkesbury river water at a dilution factor of 5. The river water was filtered through a 0.8 $\mu$m membrane prior to incubation to remove other types of suspended particulate matter such as fine minerals. The suspensions in the glass beakers were added with 5 mM NaNO$_3$ and 5 mM glucose, and were then agitated on the orbital shaker EOM5 at 100 rpm, at 20°C, under a 9:15 light:dark cycle with light being generated by a 13 W bulb in the laboratory for two weeks. This pre-incubation procedure increased the microbial population and may cause the selection of some microbial species as compared to the natural community.

### Experimental procedure and image acquisition with the OMCEC system

The OMCEC (Optical Measurement of CEll Colonisation) system was developed to overcome existing limitations in the observation and systematic measurements of suspended matter in water [32]. OMCEC can simultaneously perform material composition analysis, geometric characterisation, and motion detection of individual aggregates using different wavelength emissions, a high-quality optical system, and real-time image processing algorithms for micro Particle Tracking Velocimetry ($\mu$PTV). OMCEC can be used in combination with staining techniques on non-living materials that have similar emission spectra to biological matter [32], while blue plastic was chosen in this study to be distinguished from biological matter. OMCEC optics include a Prosilica GT3400C colour CCD equipped with a high magnification Navitar 12X Body Tube lens and a light sheet generated by optic fibers connected to a 3.7 W, 400 lumens cool white Cree LED [30]. To measure biological and MP fractions, size, shape, and settling velocity of individual MP aggregates, we used 2.75 $\mu$m per pixel CCD resolution with a field of view of 5.5 mm x 7.5 mm and camera acquisition frequency of 3.054 frames per second.

The OMCEC optical system was coupled to our fully-controlled settling column (described in [30]) that can replicate environmental hydrodynamic conditions using an adjustable turbulence generation system consisting of a scalable oscillating grid. The settling column includes a flocculation section (upper part, 210 mm x 140 mm x 600 mm) that hosts the turbulence generation system and a measuring section (lower part, 210 mm x 140 mm x 270 mm) where the OMCEC optical system resides; the two sections are separated by a diaphragm that opens during aggregate sampling (S2 Fig).

The experimental incubation and image acquisition were conducted within the settling column to minimize disturbance to the formation of microbial-associated MP aggregates. The flocculation section was filled with 8 L of filtered Hawkesbury river water added with 0.1 g L$^{-1}$ MPs, 10 mM NaNO$_3$, 10 mM glucose, and 1 L of the pre-incubated microbial sample. The settling column was loosely capped, and an 8 L of head space was available to allow sufficient gas exchange between atmosphere and the water column. The suspension was mixed for 30 minutes in every two hours at a turbulence shear rate of 20 s$^{-1}$ for 7 days to aerate the suspension and provide sufficient time for MPs and microorganisms to interact. After 7 days of mixing, MP aggregates were allowed to freely settle through a 5 mm hole in the diaphragm into the OMCEC detector, which was filled with filtered Hawkesbury river water. OMCEC acquisitions were conducted every 8 minutes and lasted 4 minutes. The grid was stopped during the $\mu$PTV acquisitions and oscillated at 20 s$^{-1}$ between acquisitions. Two frames were taken per acquisition for the motion detection.

## OMCEC image processing algorithms

Raw images from the acquisition were first systematically processed to increase brightness, remove noise, and isolate individual in-focus aggregate images (detailed in [32]). The aggregate images were then analysed to retrieve the aggregate geometrical properties as follows. The aggregate size ($L$) is calculated as the minimum square enveloping the aggregate image [35]; the projected area ($A$) is calculated as the sum of aggregate image pixels; and the outer perimeter ($P$) is the length of the aggregate outer edge. The aggregate shape was experimentally evaluated through area- and perimeter-based shape factors $b = A/L^2$, $c = P/L$, respectively. The aggregate settling velocity ($v$) is measured by OMCEC as the vertical distance travelled by an aggregate over the time difference between two consecutive frames [30].

OMCEC detects MPs and biological ballast in each aggregate by the difference in emission wavelengths between blue MPs (below 500 nm) and yellowish biological ballast (above 500 nm). Prior to the main experiment, we acquired and processed images of two control samples (i.e., MP-only and biological-only) under the same acquisition conditions and processing algorithms as for the main experiment. All pixels of the control aggregate images were scattered over the ($R_p + G_p$, $B_p + G_p$) space, with $R_p$, $G_p$, and $B_p$ being the red, green, and blue intensities of each pixel (Fig 1). The 99% confidence limits were then determined for each materials, and the bisector of the two confidence limit lines was the threshold line (Fig 1) as

$$I = 0.73(R_p + G_p) + 15.35, \tag{1}$$

where $I$ is the threshold intensity used for material classification. OMCEC assigns a pixel of an aggregate image as microplastics if its $B_p + G_p > I$, or as biological ballast if otherwise. We chose ($R_p + G_p$), and ($B_p + G_p$) as the references for segmentation because they showed the least error as compared to other channel pairs.

The biological fraction $f_b$ of an aggregate was next estimated as the number of biological pixels over the number of aggregate pixels as a first approximation of the biological content in the three-dimensional aggregate from its two-dimensional projection upon the assumption that every projection of the aggregate has statistically similar material distribution. Hence, the MP fraction was calculated as $f_{MP} = 1 - f_b$. This thresholding method resulted in 1.44% error for the MP-only sample and 4.97% error for the biological-only sample.

## Modelling of terminal velocity

The terminal velocity $\mathbf{v}^*$ of an aggregate moving in a fluid is the velocity (settling or rising) when the aggregate attains a balance of gravitational force $\mathbf{F_g}$, buoyancy force $\mathbf{F_b}$, and

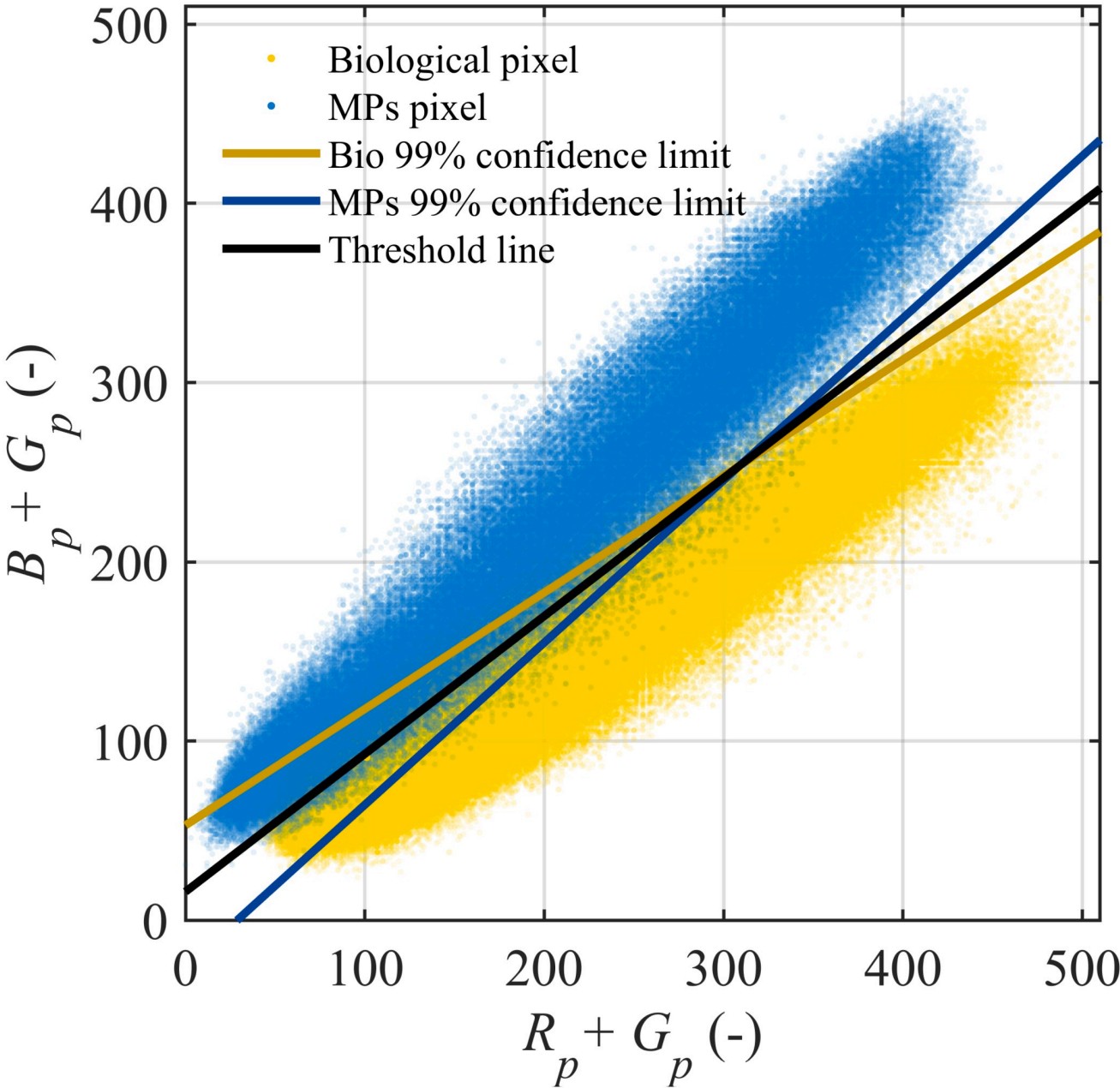

**Fig 1. Threshold graph for material segmentation.** The blue and yellow dots are pixels of MP-only and biological-only control samples, respectively. $R_p$, $G_p$, $B_p$ are the red, green, and blue intensity of a pixel, respectively.

resistance force $\mathbf{F_d}$ exerted by the fluid on the moving aggregate. The most common expression of $\mathbf{F_d}$ is through the Stoke's law [36] where $\mathbf{F_d}$ equals the viscous drag $\mathbf{F_v}$. However, the Stoke's law applies to spherical aggregates with Reynolds number $Re \ll 1$ when inertial terms are negligible and is not applicable when $Re$ increases beyond 1. More generally, $\mathbf{F_d}$ can be calculated using an empirical drag coefficient $C_d$ typically expressed as a function of $Re$ and aggregate shape (e.g., [37, 38]). To avoid empirical $C_d$, [39] and [40] used Rubey's approach [41] to model $\mathbf{v}^*$ of organic-associated, sphere-equivalent sediment aggregates considering $\mathbf{F_d} = \mathbf{F_v} + \mathbf{F_i}$, where $\mathbf{F_i}$ is the impact force exerted by fluid particle colliding onto the aggregate surface.

Following [39] and [40], we extend the use of the close-form equation for $\mathbf{v}^*$ applicable to biological-associated MP aggregates. There are possibly other forces that come into play when $\mathbf{v}^*$ increases or when the shape departs from a sphere such as the added mass inertia [42], the Basset–Bousinnesq inertia [43, 44] and others, which are not explicitly accounted for here. Therefore, the force balance describing the motion of an aggregate can be written as [39–41]

$$\mathbf{F_g} + \mathbf{F_b} + \mathbf{F_v} + \mathbf{F_i} = \hat{\mathbf{n}}_1 V\rho\|\mathbf{g}\| + \hat{\mathbf{n}}_2 V\rho_f\|\mathbf{g}\| + \hat{\mathbf{n}}_3 3\mu P\|\mathbf{v}^*\| + \hat{\mathbf{n}}_4 \frac{1}{4}A\rho_f \|\mathbf{v}^*\|^2 = 0, \qquad (2)$$

where $V$, $A$, and $P$ are the aggregate solid volume, projected area, and outer perimeter, while $\rho$, $\rho_f$, and $\mu$ are the aggregate density, fluid density, and fluid viscosity, respectively. $\hat{\mathbf{n}}_1$, $\hat{\mathbf{n}}_2$, $\hat{\mathbf{n}}_3$, and $\hat{\mathbf{n}}_4$ are unit vectors representing the directions of $\mathbf{F_g}$, $\mathbf{F_b}$, $\mathbf{F_v}$, and $\mathbf{F_i}$, respectively. $\|\mathbf{g}\|$ is the magnitude of the gravitational acceleration vector $\mathbf{g}$, and $\|\mathbf{v}^*\| = \mathbf{v}^*/\hat{\mathbf{n}}_{\mathbf{v}^*}$ is the magnitude of the terminal velocity vector $\mathbf{v}^*$ with $\hat{\mathbf{n}}_{\mathbf{v}^*}$ being the direction of $\mathbf{v}^*$. The resistance forces $\mathbf{F_v}$ and $\mathbf{F_i}$ have direction opposite to $\hat{\mathbf{n}}_{\mathbf{v}^*}$, i.e., $\hat{\mathbf{n}}_3 = \hat{\mathbf{n}}_4 = -\hat{\mathbf{n}}_{\mathbf{v}^*}$. By solving the force balance in Eq 2 for equilibrium ($\Sigma\mathbf{F} = 0$), we obtained

$$\hat{\mathbf{n}}_1 V\rho\|\mathbf{g}\| - \hat{\mathbf{n}}_1 V\rho_f\|\mathbf{g}\| - \hat{\mathbf{n}}_{\mathbf{v}^*} 3\mu P\frac{\mathbf{v}^*}{\hat{\mathbf{n}}_{\mathbf{v}^*}} - \hat{\mathbf{n}}_{\mathbf{v}^*}\frac{1}{4}A\rho_f\left(\frac{\mathbf{v}^*}{\hat{\mathbf{n}}_{\mathbf{v}^*}}\right)^2 = 0. \qquad (3)$$

Solving the quadratic Eq 3 for $\mathbf{v}^*$ results in

$$\mathbf{v}^* = 2\frac{-3\mu P \pm \sqrt{(3\mu P)^2 + \dfrac{\hat{\mathbf{n}}_1}{\hat{\mathbf{n}}_{\mathbf{v}^*}}A\rho_f\|\mathbf{g}\|(\rho - \rho_f)V}}{A\rho_f}\hat{\mathbf{n}}_{\mathbf{v}^*}. \qquad (4)$$

For a floating aggregate, $\mathbf{v}^* = 0$ implies $\rho = \rho_f$ and the acceptable solution of Eq 4 is for the positive root square. More importantly, Eq 4 can describe both the settling and rising velocity of different types of suspended aggregates depending on the aggregate density $\rho$ and fluid density $\rho_f$, where $\rho = \Sigma\rho_i f_i$ is a function of the density of each constituent material $\rho_i$ and its fraction $f_i$. An aggregate settles when $\rho > \rho_f$, floats when $\rho = \rho_f$, and rises when $\rho < \rho_f$. The sign of vector $\mathbf{v}^*$ depends on the sign of $\hat{\mathbf{n}}_1$ and $\hat{\mathbf{n}}_{\mathbf{v}^*}$. For a reference system where the vertical axis is positive downward, $\hat{\mathbf{n}}_1 = 1$, and the settling aggregates have $\hat{\mathbf{n}}_{\mathbf{v}^*} = 1$. Hence, the terminal velocity vector $\mathbf{v}^*$ for settling aggregates becomes

$$\mathbf{v}^* = 2\frac{-3\mu P + \sqrt{(3\mu P)^2 + A\rho_f\|\mathbf{g}\|(\rho - \rho_f)V}}{A\rho_f} > 0, \quad \text{sinking for } \rho > \rho_f, \qquad (5)$$

which corresponds to the equation in [39, 40]. For rising aggregates, $\hat{\mathbf{n}}_{\mathbf{v}^*} = -1$, and $\mathbf{v}^*$ becomes

$$\mathbf{v}^* = 2\frac{3\mu P - \sqrt{(3\mu P)^2 - A\rho_f\|\mathbf{g}\|(\rho - \rho_f)V}}{A\rho_f} < 0, \qquad \text{rising for } \rho < \rho_f. \qquad (6)$$

Eq 4 is valid when the Reynolds number $Re < 10$ because the original equation in [41] was shown to overestimate the drag forces when $Re \gg 10$ [45].

Here, we used Eq 4 to calculate the terminal velocity of the aggregates obtained from the main experiment with various biological fractions $f_b$. The aggregate density $\rho$ is calculated as

$$\rho = f_b\rho_b + (1 - f_b)\rho_{MP}, \qquad (7)$$

where $\rho_b$ and $\rho_{MP}$ are biological and MP densities, respectively. To lessen the number of

experimental parameters used in Eq 4, we calculate $A$ and $P$ of the aggregates as a function of $L$ exploiting the shape irregularity of the aggregates analysed later in the Results section as

$$A = b_L L^2, \tag{8}$$

$$P = c_L L, \tag{9}$$

where $b_L$ and $c_L$ are the estimated shape factors by linear fitting of experimental shape factors $b$ and $c$ against $L$ as $b_L = p_{1b}L + p_{2b}$ and $c_L = p_{1c}L + p_{2c}$ (see Results section). Aggregates observed in this study and the report in [26] suggest that microbial-associated MP aggregates possess fractal structure, which can be represented by fractal dimension $1 \leq d \leq 3$. The aggregate solid volume $V$ (i.e., excluding the pore volume $V_e = aL^3[1 - (L/L_p)^{d-3}]$) was calculated using the nonlinear fractal scaling

$$V = aL^3(L/L_p)^{(d-3)}, \tag{10}$$

where $a$ is the volume-based shape factor of the aggregate. While $a$ reflects the external shape of the aggregate as compared to a cube of the same size, the fractal dimension $d$ presents its internal structure, i.e., loose or compact. $d$ is described here after [46] as

$$d = \delta(L/L_p)^{\alpha\gamma} \quad \text{with} \quad \alpha = \begin{cases} 0, & f_b \leq 0.02 \\ 1, & f_b > 0.02, \end{cases} \tag{11}$$

where $L_p$ and $\delta$ are the size and fractal dimension of the smallest aggregate in a sample, respectively, and $\gamma$ is the fractal scaling parameter representing the rate of change in $d$ over the aggregate dimensionless size $L/L_p$. The two-parameter Heaviside function is applied in Eq 11 to take into account the compact internal structure of virgin MPs (i.e., $f_b \leq 0.02$ in this study) with fractal dimension $d \approx 3$ regardless of size $L$.

Parameters $a$, $\gamma$, and $\rho_b$ in Eq 4 were unknown and thus were estimated against the experimental settling velocity $v$ using PEST [47] (see Results section).

### Ethics statement

The sampling of natural microorganisms and river water did not require permit because it was conducted in a public access section of the river. We confirm that the field study did not involve endangered or protected species.

## Results

### Biological colonisation on MP aggregates

Our experimental data include 702 high-resolution images of irregularly-shaped polyurethane MP aggregates colonised by microorganisms (see examples in Fig 2 and S1b Fig). We retrieved the biological fraction ($f_b$ between 0 and 1) using the OMCEC system to quantify the microbial colonisation level. Observations spanned nearly the entire $f_b$ range. Biological matter attached only locally on the aggregate surfaces when $f_b$ was low but spread more widely at higher $f_b$. At intermediate to high $f_b$ values, biological matter also bridged two or more MPs and gradually covered the MP outer surface.

### Effects of biological colonisation on MP aggregate size and shape

**MP aggregate size.** The size of virgin MP particles obtained from the MP-only sample ranged from 50 to 500 $\mu$m (grey line in Fig 3a), while we physically sieved MPs to the size of 90

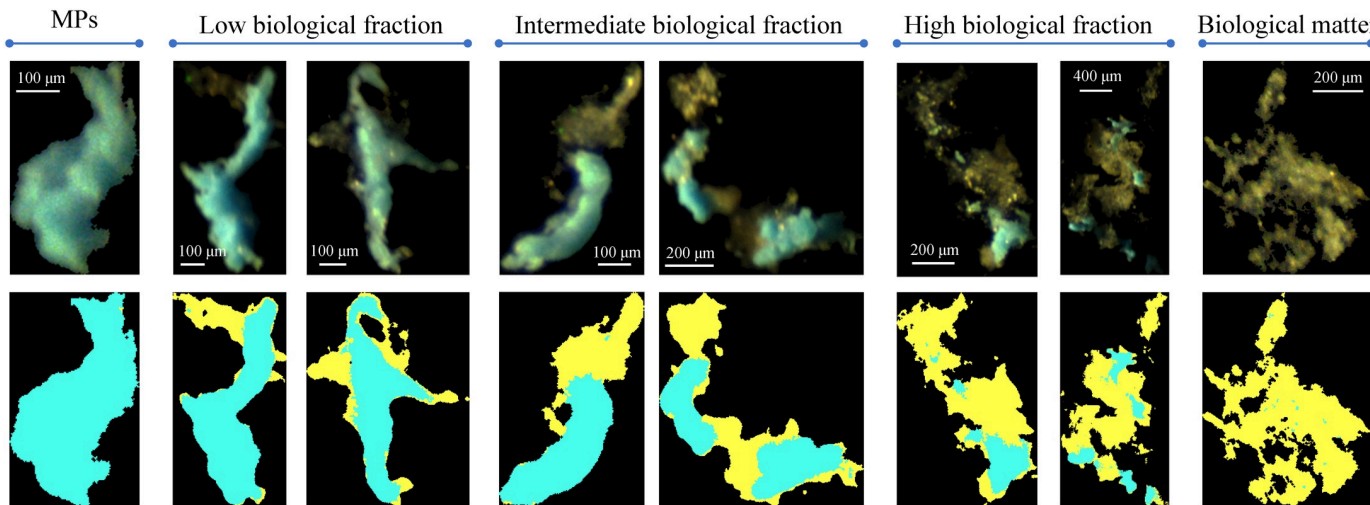

**Fig 2. Samples of microbial-associated suspended MP aggregates acquired with OMCEC.** The fraction of attached biological ballast increases from left to right. Top row: raw images. Bottom row: segmented images with blue representing MPs and yellow representing biological matter. Note that the aggregate size varies among the aggregate images as shown in the scale bar in each image.

to 300 $\mu$m prior to the experiments. We attribute these differences to the shape of the MP particles (e.g., elongated) that allowed MPs with size > 300 $\mu$m to pass through the square mesh, and the tiny dust that may stick on other particles during the sieving process. Interestingly, MP aggregates containing less than 10% biological ballast had size remarkably larger than virgin MP particles (i.e., double the minimum and maximum size).

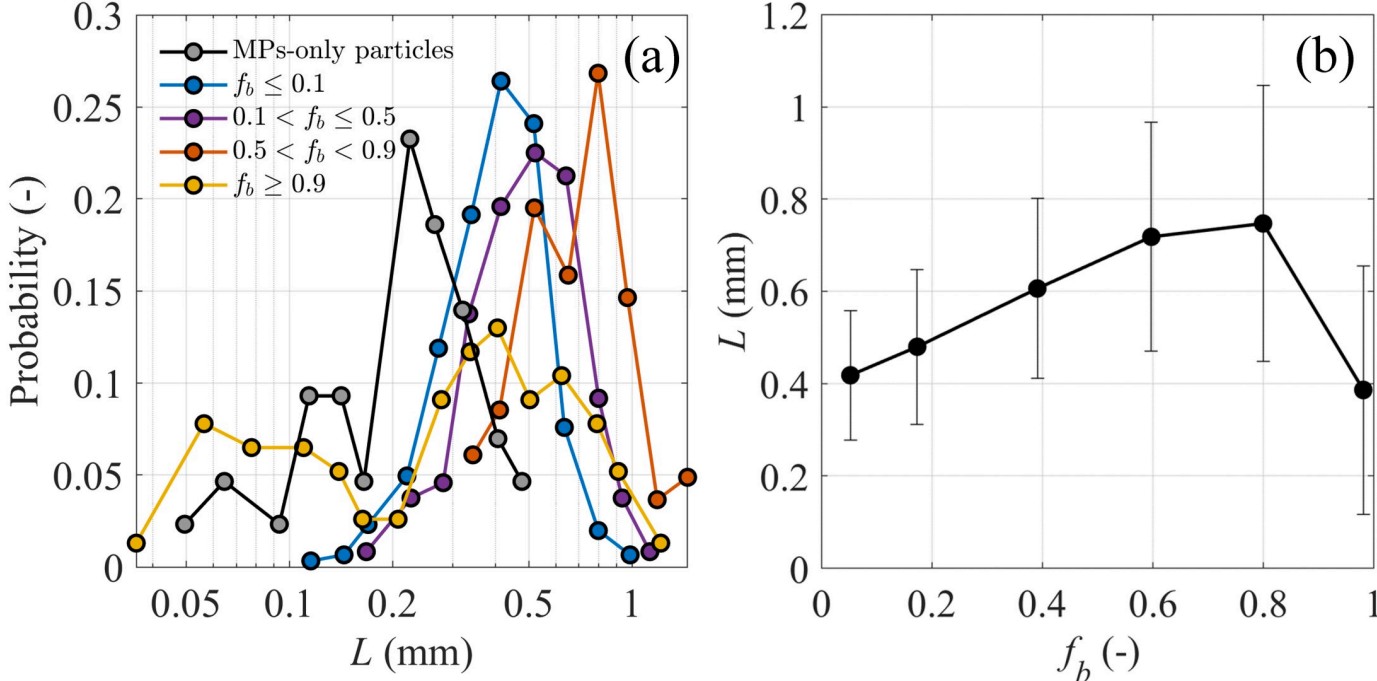

**Fig 3. MP aggregate size analyses.** (a) Probability distribution of size $L$ for microbial-associated MP aggregates at different biological fraction $f_b$ and virgin MP particles in the MP-only sample. (b) Experimental bin-averaged size $L$ of microbial-associated MP aggregates as a function of the attached biological fraction $f_b$; error bars represent standard deviation in each bin.

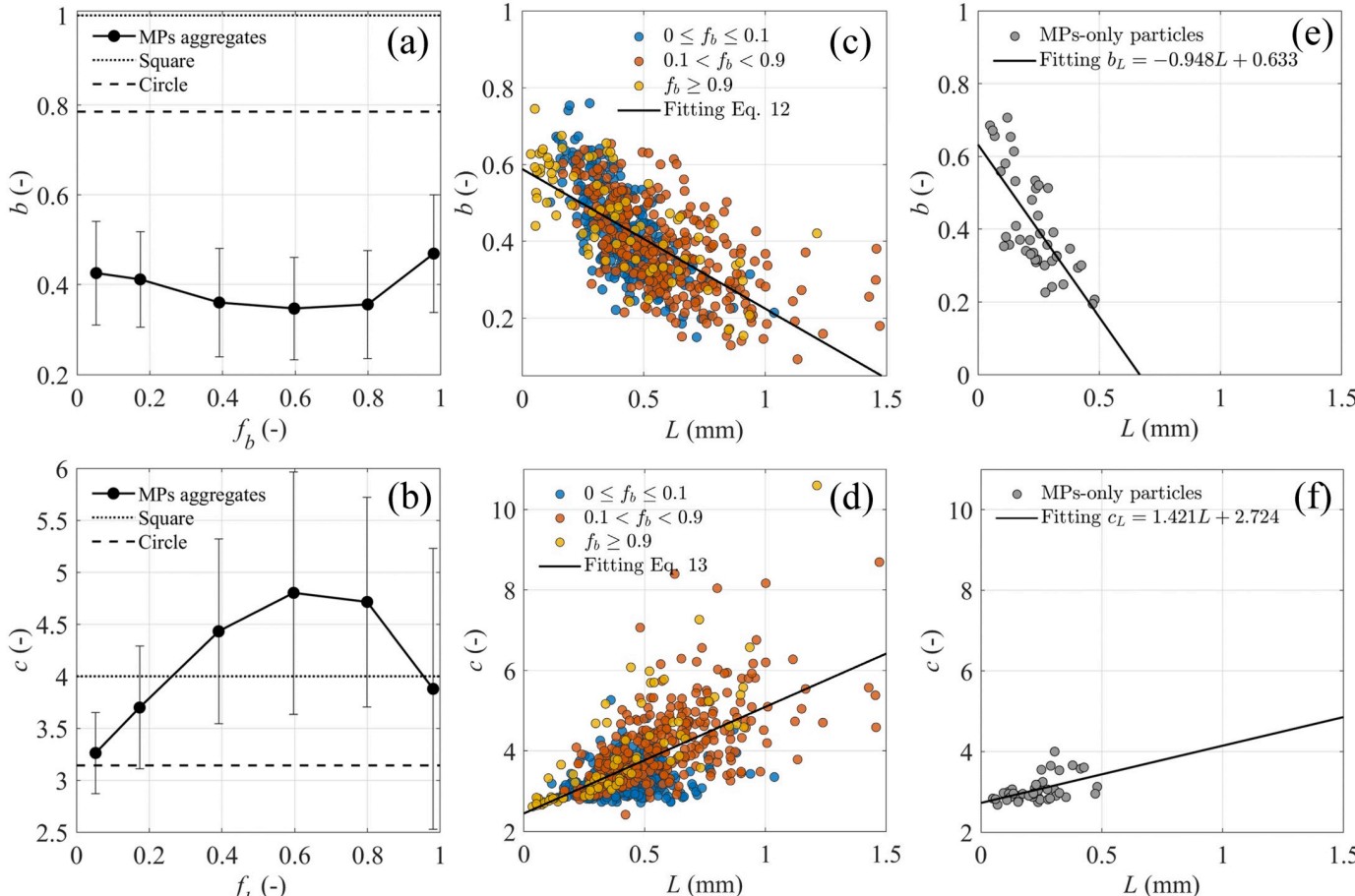

**Fig 4. MP aggregate shape analyses.** Experimental bin-averaged (a) area-based shape factor $b$, and (b) perimeter-based shape factor $c$ of microbial-associated MP aggregates as a function of the attached biological fraction $f_b$; error bars represent standard deviation in each bin. (c) and (d) scatter of the shape factors $b$ and $c$ of microbial-associated MP aggregates over aggregate size $L$ for three ranges of $f_b$ from the main experiment. (e) and (f) scatter of the shape factors $b$ and $c$ of virgin MP particles over particle size $L$ from the MP-only sample. $b_L$ and $c_L$ were the shape factors calculated as a function of $L$ as in Eqs 12 and 13.

The size distribution of MP aggregates shifted towards larger $L$ with increasing biological fraction $f_b$, except for aggregates with $f_b \geq 0.9$ (Fig 3a). This finding agrees with earlier literature showing larger aggregates due to the growth of living microorganisms and their secretion of extracellular polymeric substance [30, 48]. The broad span of the size distribution of the biologically-prevalent group ($f_b \geq 0.9$) is likely because it included aggregates with different characteristics such as pure biological aggregates over a wide $L$ range as well as MP aggregates at high levels of biological colonisation. In contrast to our expectation that the size of MP aggregates correlates positively with biological fraction, the maximum $L$ observed experimentally was found at $f_b \simeq 0.8$ (Fig 3b). A possible explanation is that aggregates broke up at high $f_b$ due to weaker biological bridges between particles or negative effects of microbial activity such as grazing and decomposition [48, 49].

**MP aggregate shape.** The experimentally-retrieved shape factors of MP aggregates in this experiment had projected areas and perimeters largely departing from regular spheres ($b = \pi/4$, $c = \pi$, Fig 4). MP aggregates occupied less than half the area of the containing square, and the MP aggregates with $0.4 \leq f_b \leq 0.8$ occupied the least area (Fig 4a). In contrast, these

MP aggregates had the longest outer perimeter (i.e, $c > 4$, Fig 4b). Therefore, microbial-associated MP aggregates have more irregular shapes than aggregates with one prevalent material.

The aggregate shapes were not only affected by $f_b$ but also changed with $L$ (Fig 4c and 4d). The larger the MP aggregates grew, the less area they filled (Fig 4c) and the more irregular surfaces they had (Fig 4d). The experimental shape factors $b$ and $c$ were expressed as linear functions of $L$ as

$$b_L = -0.364\,[\text{mm}^{-1}]L + 0.588, \qquad (12)$$

$$c_L = 2.649\,[\text{mm}^{-1}]L + 2.448, \qquad (13)$$

where $L$ is in mm. Virgin MP particles from the MP-only sample also had irregular shapes (Fig 4e and 4f), which were caused by the sawing production and replicated the MP particles found in natural waters that formed as a result of biological and chemical degradation as well as physical fragmentation [50].

## Effects of biological colonisation on MP aggregate settling

Our experiment showed a 50% decrease in the settling velocity $v$ of MP aggregates when $f_b$ increased from 0.05 to 0.8 (Fig 5a). The most susceptible size range to biological colonisation was for $L \geq 0.7$ mm, where we found the highest rate of change in $v$ per incremental change in $f_b$. Although almost all changes were negative, that is, $v$ decreased with increasing $f_b$, the rate of change at medium to high $f_b$ was more apparent than that at low $f_b$.

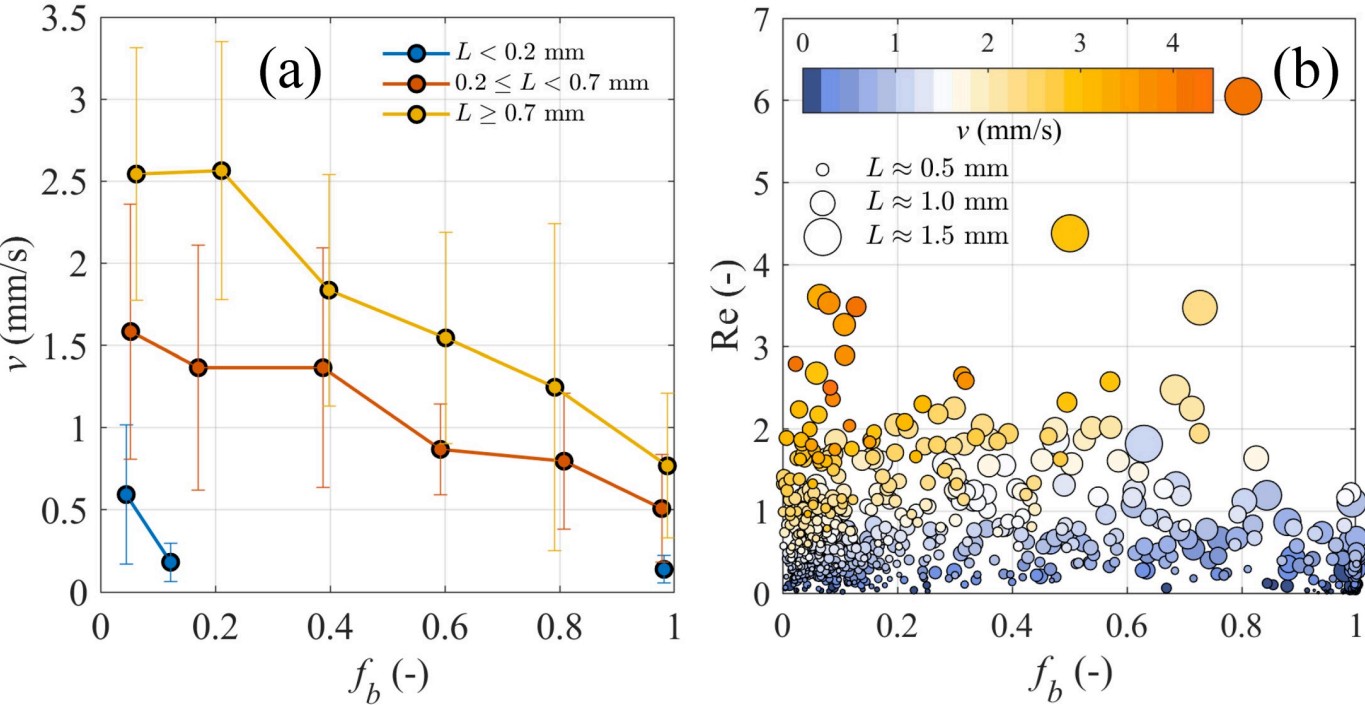

**Fig 5. MP aggregate experimental settling velocity analyses.** (a) Bin-averaged experimental settling velocity $v$ for aggregates at different size $L$ as a function of biological fraction $f_b$; error bars represent standard deviation in each bin. (b) Reynolds number Re over ranges of $f_b$; colours of scattered points represent values of $v$ and size of scattered points is linearly scaled with $L$ ranging from 0 to 1.5 mm.

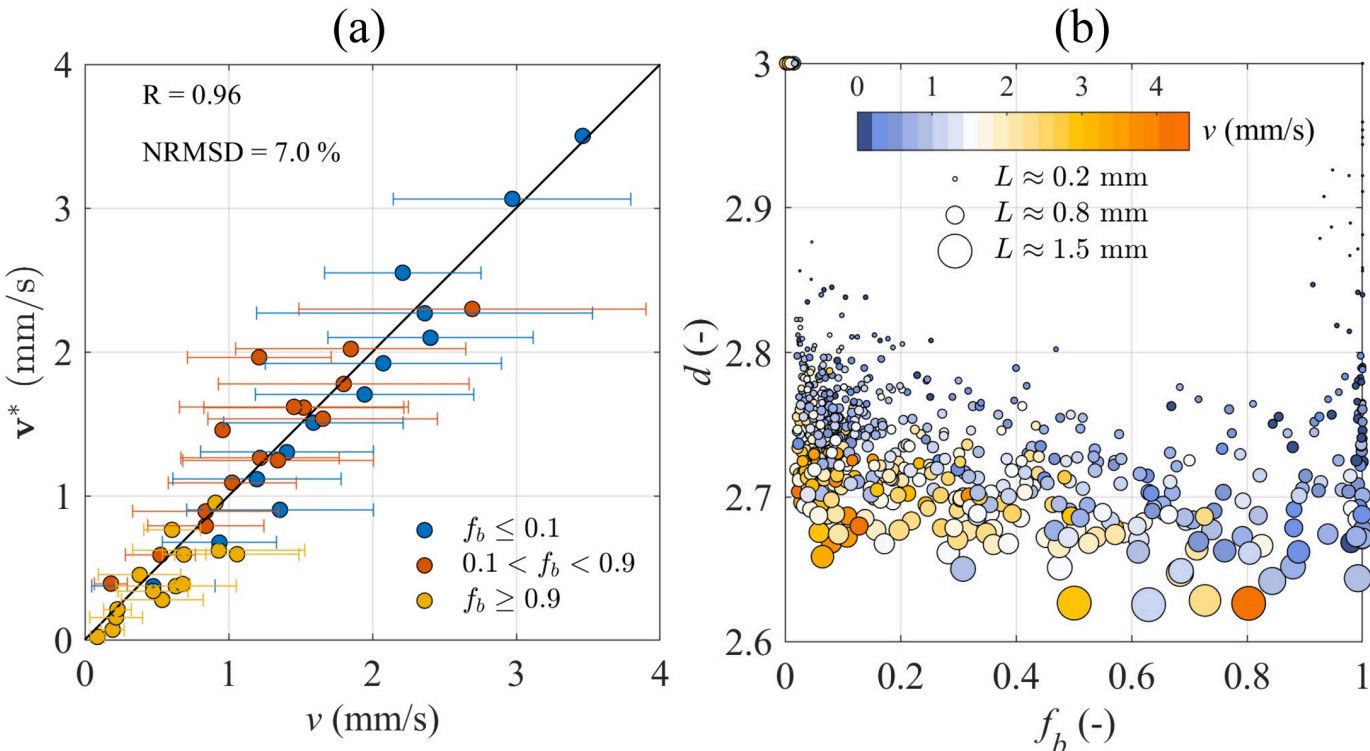

**Fig 6. Terminal velocity equation validation and aggregate fractal dimension estimation.** (a) Validation of the terminal velocity Eq 4 for microbial-associated MP aggregates; $v$ is the experimental settling velocity while $\mathbf{v}^*$ is the estimated settling velocity using Eq 5. (b) Estimated aggregate fractal dimension $d$ over ranges of experimental aggregate size $L$, biological fraction $f_b$, and settling velocity $v$, respectively.

To understand the settling regime of MP aggregates, we calculated $Re = \rho v L/\mu$ with the experimental $L$ and settling velocity $v$, and with density $\rho$ estimated using Eq 7. MP aggregates in our experiments had $10^{-3} < Re < 6.1$ (Fig 5b), which reached the upper bound of the Stokes regime but were within the limit of validity for Eq 4. The experimental points in Fig 5b show that high $Re$ corresponded to large $L$ and high $v$, and vice versa. In addition, although $f_b$ was accounted for in the calculation of $Re$ through $\rho$, its contribution was weak and we could not detect a trend between $Re$ and $f_b$ (Fig 5b).

## Modelling of the settling velocity of MP aggregates

The terminal velocity $\mathbf{v}^*$ in Eq 5 was used with our experimental data (Fig 6a). The smallest aggregate in the experiment had $L_p = 37.5 \,\mu m$ and fractal dimension assumed to be $\delta = 3$; the experimental shape factors in Eq 4 were substituted by the $L$-dependent $b_L$ and $c_L$ values in Eqs 12 and 13. The MP density $\rho_{MP} = 1.195 \text{ g cm}^{-3}$ was obtained from the manufacturer data sheet of PUR pipe (RS. 483-5765), while the experimental density and viscosity of Hawkesbury river water were $\rho_f = 1.007 \text{ g cm}^{-3}$ and $\mu = 1.002 \text{ mPa·s}$ at 20°C, respectively. The volume-based shape factor $a$, fractal scaling parameter $\gamma$, and biological density $\rho_b$ were unknown and were therefore estimated against the experimental $v$ (in brackets in Table 1) by inverse problem solution of Eq 5 using a nonlinear least-square fitting until the residuals between the observed and calculated velocities were the minimum [47]. Here, we discretized the $L$ range into three groups by $f_b$ values to optimize the least-square fitting minimization.

The estimated settling velocity $\mathbf{v}^*$ was shown to match the experimental $v$ well (with correlation coefficient R = 0.96 and the normalized root-mean-square deviation NRMSD = 7%,

**Table 1. Parameters used for settling velocity in Eq 5.** Parameters in brackets were estimated from calibration, while the others were measured from experiments, assumed, and retrieved from the manufacture data sheet.

| Parameters | Units | Values | Definition |
|---|---|---|---|
| $v^*$ | [mm s$^{-1}$] | 0.019 to 3.501 | Modelled terminal velocity, Eqs 4,5,6 |
| $\mu$ | [mPa·s] | 1.002 | Water viscosity at 20˚C |
| $\rho_f$ | [g cm$^{-3}$] | 1.007 | Water density, measured for Hawkesbury River water |
| $\rho$ | [g cm$^{-3}$] | 1.048 to 1.195 | Aggregate density, Eq 7 |
| $\rho_b$ | [g cm$^{-3}$] | (1.048) | Biological matter density, calibrated |
| $\rho_{MP}$ | [g cm$^{-3}$] | 1.195 | Polyurethane (PUR) density, from manufacture data sheet |
| $\|g\|$ | [m s$^{-2}$] | 9.81 | Gravitational acceleration |
| $L$ | [mm] | 0.0375 to 1.471 | Experimental aggregate size |
| $A$ | [mm$^2$] | 0.00073 to 0.227 | Aggregate projected area, calculated from the shape factor $b_L$ and size $L$, Eq 8 |
| $P$ | [mm] | 0.091 to 9.336 | Aggregate outer perimeter, calculated from the shape factor $c_L$ and size $L$, Eq 9 |
| $b_L$ | [-] | 0.053 to 0.575 | Estimated area-based shape factor from the experimental shape factor $b$ against $L$, Eq 12 |
| $c_L$ | [-] | 2.543 to 6.345 | Estimated perimeter-based shape factor from the experimental shape factor $c$ against $L$, Eq 13 |
| $p_{1b}, p_{1c}$ | [mm$^{-1}$] | -0.364, 2.649 | Linearly fitted parameters in Eq 12 |
| $p_{2b}, p_{2c}$ | [-] | 0.588, 2.448 | Linearly fitted parameters in Eq 13 |
| $V$ | [mm$^3$] | 4.2x10$^{-6}$ to 0.076 | Aggregate solid volume (excluded pores), Eq 10 |
| $a$ | [-] | (0.093) | Volume-based shape factor, calibrated |
| $d$ | [-] | 2.630 to 3 | Aggregate fractal dimension, Eq 11 |
| $\delta$ | [-] | 3 | Fractal dimension of the smallest aggregate in the experiment, assumed |
| $\gamma$ | [-] | (-0.0359) | Fractal scaling parameter, calibrated |
| $L_p$ | [mm] | 0.0375 | Size of the smallest aggregate in the experiment, measured |
| $\alpha$ | [-] | 0, 1 | Heaviside parameter in Eq 11 taking into account the compact structure of virgin MPs regardless of size $L$ |

Fig 6a). The estimated volume-based shape factor $a$ = 0.093 suggested that the aggregates in average occupied less than 10% of the volume of a cube of the same size. The estimated fractal scaling parameter $\gamma$ = −0.0359 has commonly been found for biological aggregates [40] and depicted a negative relationship between $L$ and $d$. The biological density was estimated as $\rho_b$ = 1.048 g cm$^{-3}$, which was within the range of $\rho_b$ of diatoms used in [51–53] and estimated in [40].

## Effects of biological colonisation on MP aggregate fractal dimension

Using the estimated fractal scaling parameter $\gamma$ = −0.0359, we were able to calculate MP aggregate fractal dimension $d$ as in Eq 11 and represent it against $f_b$, $v$, and $L$ (Fig 6b). MP aggregates had $d$ ranging from 2.6 to 3 with minimum $d$ at intermediate to high $f_b$, suggesting that MPs colonised by biological matter in our experiments resulted in less compact aggregate architecture as compared to MP-prevalent ($f_b \leq 0.1$) and biologically-prevalent ($f_b \geq 0.9$) aggregates. $d$ was positively correlated to $L$ (see marker size in Fig 6b), but an apparent correlation between $d$ and the experimental $v$ was not observed (see marker colours in Fig 6b) although the parameter $\gamma$ used to calculate $d$ was estimated against $v$. It is possible to conclude that $d$ may indirectly impact $v$ through its effects on $L$ and $f_b$, and that the vertical transport of MP aggregates is non-linearly controlled by a combination of correlated parameters including size, shape, fractal dimension, and material fractions.

## Modelled terminal velocity of low- and high-density MPs

Eq 4 was applied to model the terminal velocity (settling or rising) of seven different common polymers with two types of biological ballast (Fig 7) at different $f_b$ levels. The analysis was

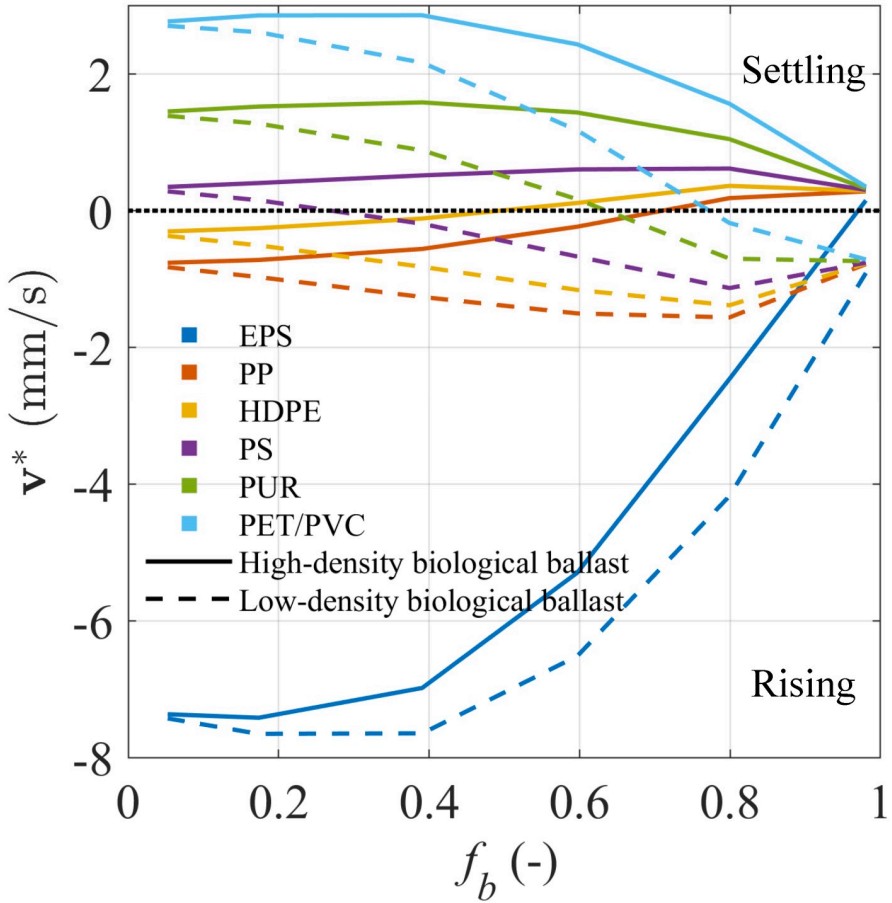

**Fig 7. Analyses of low- and high-density MPs associated with biological ballast.** Estimated terminal velocity $\mathbf{v}^*$ of microbial-associated MP aggregates made of seven different polymers and two types of biological ballast against the biological fraction $f_b$ range. Low-density polymers include expanded polystyrene (EPS), polypropylene (PP), and high-density polyethylene (HDPE), and high-density polymers include polystyrene (PS), polyurethane (PUR), polyethylene terephthalate (PET), and polyvinyl chloride (PVC).

conducted using the average MP aggregate geometrical properties observed in our experiment for polyurethane with same values applied to all other polymers upon the assumption that microbial colonisation patterns on different polymers are similar. The low-density polymers included expanded polystyrene (EPS, $\rho_{MP} = 0.011$ g cm$^{-3}$), polypropylene (PP, $\rho_{MP} = 0.905$ g cm$^{-3}$), and high-density polyethylene (HDPE, $\rho_{MP} = 0.965$ g cm$^{-3}$), and the high-density polymers included polystyrene (PS, $\rho_{MP} = 1.050$ g cm$^{-3}$), polyurethane (PUR, $\rho_{MP} = 1.195$ g cm$^{-3}$), and polyethylene terephthalate (PET) and polyvinyl chloride (PVC) with similar densities $\rho_{MP} = 1.370$ g cm$^{-3}$. $\rho_b = 1.048$ g cm$^{-3}$ in our experiment was used to represent the high-density ballast found in natural waters, and $\rho_b = 0.900$ g cm$^{-3}$ was used to represent ballast mainly made of algae and cyanobacteria with very low density or entrapping gas bubbles [54].

Microbial colonisation can cause MP aggregates to either retain their motion direction and change only their speed or switch the motion direction (Fig 7). PVC, PET, and PUR aggregates reduced by more than a half their settling velocities when attached to high-density ballast, while PP and HDPE aggregates almost doubled their rising velocities when attached to low-density ballast. On the other hand, the naturally settling MP aggregates (PS, PUR, PET and PVC) became neutral-buoyant and rising with 30 to 80% of low-density ballast, and the

naturally rising MP aggregates (HDPE and PP) started to settle with 50% of high-density ballast. PS aggregates attached to high-density ballast with $\rho_{MP} \simeq \rho_b$ tended to sink faster due to the dominant effect of the increased size on the settling velocity. In contrast, EPS, a polymer with very low density, decreased its rising velocity when being colonised but almost never settled.

## Discussion

Microbial-associated MP aggregates have been recognised and studied previously but this is the first work that investigates and quantifies the attached biological fraction and its effects on the geometrical properties and terminal velocity of MP aggregates. The attached biological fraction has a well defined linear impact on the MP aggregate density $\rho$ (see Eq 7): for any MP density, the biological ballast can bring $\rho$ close to the ambient fluid density $\rho_f$, or cause $\rho$ to exceed or fall below $\rho_f$. However, the terminal velocity of suspended MP aggregates is controlled not only by their excess density but also by their architecture such as size, shape, and fractal dimension [40]. Our experiment and analyses showed that the correlation between $f_b$ and MP aggregates architecture was nonlinear, with MP aggregates containing about 60% to 80% of biological ballast having the largest size, the most irregular shape, and the lowest fractal dimension. These counter-balancing effects caused MP aggregates terminal velocity to be nonlinearly affected by $f_b$, where a higher rate of change in $v$ per unit change in $f_b$ was observed at high $f_b$ values.

In natural waters, microbial-associated MP aggregates can be formed by many different mechanisms. Microorganisms can attach to MP surfaces by means of their motility and by turbulence, and grow colonies on MPs through metabolic growth and excretion of sticky byproducts [55, 56]. MPs can also incorporate into pre-formed biological aggregates or marine snow by the aid of turbulence vorticity as shown in [26, 29]. The formation of microbial-associated MP aggregates in our experiment was likely related to both mechanisms. From the aggregate images (in addition to those in Fig 2), we observed the spread of biological ballast on MP surface, which was possibly caused by cell colonisation, and we also observed MPs embedded in large biological aggregates, which was a sign of incorporation. Bio-fouling on MPs [28] and the degradation and fragmentation of pre-colonised macroplastics can also produce microbial-associated MP aggregates. Especially, MPs can be associated with biological ballast as a product of trophic transfer for example when MPs are ingested and then egested in faecal pellets by zooplankton [27].

In all previous experiments [26–29, 55], attached biological ballast was shown to activate or increase the MP settling velocity leading to fast deposition to the bed. Our experiment shows evidence of a decreasing $v$ of high-density MP aggregates following microbial colonisation, which is explained by biological ballast having density $\rho_b = 1.048$ g cm$^{-3}$ (i.e., retrieved from the settling velocity equation calibration, Table 1) lower than the MP density $\rho_{MP} = 1.195$ g cm$^{-3}$. This finding is important because, first, the microorganisms we cultured were sampled from a natural brackish river from the suburban Sydney area, and thus they represent the microbial composition mostly exposed to MP pollution in this large urban settlement; and, second, the tested polyurethane MPs belong to the most common high-density polymers. Given that the residence time $t_r$ of MP aggregates in the water column scales with $v$ as $t_r \propto 1/v$ (excluding other factors like turbulent and Stoke drifts), it is likely that MP presence within the water column of the sampled environment is increased by microbial ballast.

The proposed Eq 4 captured the experimental velocity of microbial-associated MP aggregates well. Eq 4 was developed in [39] for organic-associated sediment aggregates, and has been adapted here to account for virgin MPs and microbial-associated MP aggregates. Except

for MPs with $Re \ll 1$ that may still follow the Stokes regime [5, 57], the settling velocity of virgin MPs with $Re > 1$ (i.e., non-negligible inertia force) has previously been estimated using empirical definitions of drag coefficients $C_d$ and various particle sizes and shapes. For example, [58, 59] employed the equation in [37] for regular and irregular virgin MPs of different polymers, while [60] suggested the use of the equation in [61] for spherical and round-like MPs of various polymers. However, MPs associated with biological matter are characterized by fractal structure as shown in [26] and in this work, and thus have remarkably different size, shape, and terminal velocity as compared to virgin MP particles. In addition, Eq 4 can account for changes in the direction of motion, that is from settling to rising and vice versa, as disscused below.

By means of the OMCEC system and the terminal velocity Eq 4, we quantitatively showed that the association of biological ballast can induce a switch in the direction of MP terminal velocity, and this may explain the presence of low-density MPs (e.g., polyethylene-propylene, PE, PP) at the ocean bed [62–65] and high-density MPs (e.g., PS, PVC, PET) at the water surface [66], as well as the massive loss of MPs from the water surface [67]. After switching the motion direction and before reaching the ocean bed or surface, MP aggregates sink or rise at a lower speed as compared to other virgin MPs moving in the same direction. Even if MPs retain their motion direction, their terminal velocity can also be substantially reduced due to biological ballast association. Consequently, the residence time of both high- and low-density MPs at different microbial colonisation levels would be lengthened and MP presence would stretch along the water column for longer period. Aquatic organisms at different depth of the water column, therefore, may have more time and chance to encounter MPs.

This research was conducted with the acknowledgment of some technical limitations and assumptions. The first limitation was the aggregate size limit that can be detected by OMCEC optics. Although the resolution was 2.75 $\mu m$ per pixel, the smallest aggregate size we reported was 37.5 $\mu m$ to minimize the errors due to noise and light reflection. Large aggregates exceeding our field of view or sinking faster than our camera speed were also not captured. The second limitation was the two-dimensional image acquisition with an assumption of similarities of aggregate properties on every projection. However, given the large number of samples and the stochastic nature of aggregate structure, results were considered to be statistically reliable and repeatable by other imaging systems such as In Situ Settling Velocity (INSSEV) and Laser In-Situ Scattering and Transmissometery (LISST) [68–70] or laboratory-based systems [71–73]. Thirdly, natural water column consists of a wide variety of suspended materials, especially minerals, and thus MP aggregates are more likely to contain as well minerals, detritus, and faecal pellets that can significantly affect their vertical transport through changes in density, size, and fractal dimension. Fourthly, the use of only polyurethane MPs to represent other MPs originated from different polymers is only a proximity estimation because polymers have distinct physical and chemical properties. MPs of different polymers may be generated with different sizes and shapes even under similar manufacturing methods. The differences in their polymer formulation, bonds, chemical groups, and added additives can favour the microbial activities differently, and thus the microbial colonization quantities and patterns on different polymers are very likely to be different. For example, PUR containing nitrogen and having flexible bonds is a favourable substrate for biodegradation [74, 75]. Moreover, microorganisms with diverse metabolisms and colonization preferences [54] are very likely to produce different colonization patterns on MPs. MP aggregates formed under various environmental conditions (e.g., shear rates, nutrient and particle concentrations) with different cell colonization patterns may variegate in geometrical properties and velocities, as suggested by studies of microbial-associated mineral aggregates [30, 32, 40]. Therefore, more experiments on specific polymers

and microorganisms, as well as their complex interactions in natural environments, are needed to provide a better understanding of MP transport in natural waters.

## Supporting information

**S1 Fig. Example microscope images.** (a) Microorganisms sampled at Hawkesbury River, NSW, Australia after 2 weeks of pre-incubation, and (b) an aggregate of blue polyurethane microplastics and microorganisms from (a). Images were acquired using a Motic® series BA310 microscope at 10X magnification.
(TIF)

**S2 Fig. Settling column.** (1a) flocculation section, (1b) diaphragm, (1c) measuring section, (2) oscillating grid, (3) DC motor that drives the grid, (4) water quality meter, (5) Cree LED, (6) optical fibres, (7) camera and magnification lens and (8) microcontrolling system. (5), (6), and (7) together make the OMCEC optical system. Figure was credit from [30].
(TIF)

**S1 Table. Experimental size, biological fraction, and settling velocity of MP aggregates.**
(XLSX)

## Acknowledgments

We thank Peter Scanes, Jaimie Potts, and Angus Ferguson, New South Wales Office of Environment and Heritage (NSW-OEH), for providing assistance in collecting samples in the Hawkesbury river.

## Author Contributions

**Conceptualization:** Thu Ha Nguyen, Fiona H. M. Tang, Federico Maggi.

**Data curation:** Thu Ha Nguyen.

**Formal analysis:** Thu Ha Nguyen.

**Funding acquisition:** Federico Maggi.

**Investigation:** Thu Ha Nguyen, Fiona H. M. Tang.

**Methodology:** Thu Ha Nguyen, Fiona H. M. Tang, Federico Maggi.

**Resources:** Thu Ha Nguyen.

**Software:** Thu Ha Nguyen.

**Supervision:** Fiona H. M. Tang, Federico Maggi.

**Validation:** Thu Ha Nguyen.

**Visualization:** Thu Ha Nguyen.

**Writing – original draft:** Thu Ha Nguyen.

**Writing – review & editing:** Thu Ha Nguyen, Fiona H. M. Tang, Federico Maggi.

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
