## [Decision Letter · Decision Letter 0]

4 Dec 2019

PONE-D-19-30091

Sinking of microbial-associated microplastics in natural waters

PLOS ONE

Dear Mrs Nguyen,

Thank you for submitting your manuscript to PLOS ONE. After careful consideration, we feel that it has merit but does not fully meet PLOS ONE’s publication criteria as it currently stands. Therefore, we invite you to submit a revised version of the manuscript that addresses the points raised during the review process.

We would appreciate receiving your revised manuscript by Jan 18 2020 11:59PM. To enhance the reproducibility of your results, we recommend that if applicable you deposit your laboratory protocols in protocols.io, where a protocol can be assigned its own identifier (DOI) such that it can be cited independently in the future. For instructions see: http://journals.plos.org/plosone/s/submission-guidelines#loc-laboratory-protocols

We look forward to receiving your revised manuscript.

Kind regards,

Matthäus Bäbler, Ph.D.

Academic Editor

PLOS ONE

Journal Requirements:

1. Our internal editors have looked over your manuscript and determined that it may be within the scope of our Plastics in the Environment Call for Papers. The Collection will encompass a diverse range of research articles to better understand various aspects of the effect of plastics in the environment. Additional information can be found on our announcement page: https://collections.plos.org/s/plastics-environment. If you would like your manuscript to be considered for this collection, please let us know in your cover letter and we will ensure that your paper is treated as if you were responding to this call. If you would prefer to remove your manuscript from collection consideration, please specify this in the cover letter.

2. In your Methods section, please provide additional location information, including geographic coordinates for the data set if available.

Reviewers' comments:

Reviewer's Responses to Questions

**Comments to the Author**

1. Is the manuscript technically sound, and do the data support the conclusions?

Reviewer #1: Partly

Reviewer #2: Yes

2. Has the statistical analysis been performed appropriately and rigorously? 

Reviewer #1: Yes

Reviewer #2: Yes

3. Have the authors made all data underlying the findings in their manuscript fully available?

Reviewer #1: Yes

Reviewer #2: Yes

4. Is the manuscript presented in an intelligible fashion and written in standard English?

Reviewer #1: Yes

Reviewer #2: Yes

5. Review Comments to the Author

Reviewer #1: The manuscript deals with an original and intelligent approach to predict the behavior of different microplastic particles associated with different degrees of biological colonization in natural water. The experimental database is very well explained and carefully evaluated. I have to admit that I did not check the equations, but their construction has been explained clearly and understandable for non-specialists. It should be stated more clearly in the text where the conclusions are based on the authors’ own experiments as opposed to theoretical simulation. In the latter case, some assumptions need critical evaluation (see below). To my opinion there is already some discussion of findings in the results section, which should be commented by the handling editor.

Abstract:

line 3 should probably read „excess buoyancy relative to water“

“MPs” in plural form is sometimes incorrect, think of using “MP” throughout

Last phrase: statement should be expressed more cautiously, as the application is purely theoretical and has limitations (see later)

Lines 2-3: statement is misleading, the references point to input from rivers, which integrate many pathways of plastic input, among which household sewage systems do not dominate

Lines 17-18: plastic does not feed higher trophic levels (not even the level where it enters the food chain) – better: it can be transferred to higher trophic levels

Line 84: “cycle” instead of “cycles” – please also give details on the vessel shape and size, the light intensity and incubator type

Line 228: up to 9.x % biological ballast is not “without biological matter”, this only fits for the virgin plastic particles – please adjust statement

Line 309: it should be said more clearly that this part of the study is mostly based on literature data, not on experimentation.

Line 336: I do not understand how this number is derived from Eriksen et al.’s study, since they reported particle counts and weights on a square kilometer basis. Moreover, the comparison is critical, as Eriksen et al. recorded particles larger than 300 µm, and the study under review used PUR sawdust sized 90-300 µm. Please explain more clearly.

Lines 380-381: please say more clearly that only PUR has been experimentally tested. It may differ in colonization from other polymers due to several factors, one of which is the fact that it is the only polymer containing nitrogen (see e.g. study by Russell et al. 2011, https://aem.asm.org/content/77/17/6076)

Line 411: please check carefully if the given weight is correct. Reference 69 includes plastic of different size classes, and reference 70 reports plastic particles larger than 1 cm throughout. Moreover, even if the particles sink faster, this does not necessarily mean that fish do not catch them, they might be attracted by their motion.

Line 415 and thereafter: regarding the limitations of the study, it should also be considered that the shape characteristics of the other polymers might be different even after using the same sawing procedure due to their physical properties. This also applies to their wettability, which may greatly influence colonization and transport.

Line 428: faecal or fecal pellets

Reviewer #2: The manuscript „Sinking of microbial-associated microplastics in natural waters” by Nguyen et al. is original, interesting, timely, and an important contribution to the current knowledge of marine plastic research. The experimental design is elegant and the results provide new insight into the movement behavior of microplastic within the water column. The combination of lab experiments and model approaches expand the knowledge about the movement of different plastic types in the water column in relation to biofouling. The only criticism I have is the way the results are presented. The authors presume that the readers are experts in the research field of the authors. Important explanatory details about the methods are missing. This makes the ms very difficult to read and to understand. The authors used a lot of different parameters, some of them not properly explained, which can easily confuse the reader. I recommend to add more explanation and rewrite some parts of the ms in a simpler way. The paper will gain more visibility when it is easy to understand. I hope my comments below can help to focus on parts that have room for improvements.

Major comments:

You present a lot of calculations in your ms with plenty of parameters. Most of the parameters are explained somewhere in the text. However, it is easier to follow the text with having an overview of all parameters. Can you expand Table 1 and include all parameters you used in your equations? Explain in an additional column in the same table the meaning of the parameters. Something like:

Parameter | Units | Values | definition | equation

A | µm | 10-200 | projected area = sum of image pixels of an aggregate | 1, 2, 3,…

Line 66-67: Please explain in more details in the ms what do you mean with “shape irregularity and fractal structure”.

Line 75: You used a plankton net with 20µm mesh size and you said that you sampled bacteria (line81). Most bacteria range between 0.2 and 2µm. Do you wanted to exclude most of the bacteria? Why you didn’t use unfiltered water for your pre-incubation to obtain the entire biota? And why do you pre-incubated the water before you bring it in the OMCEC system? Please explain in the ms.

Line 85: You added nitrate and glucose to your experiment, why? Do you want to trigger a phytoplankton bloom? Please explain in the ms.

In general, can you give more details in your ms why you manipulated the plankton community before you used them in the OMCEC system? Is your community still a natural community or is there a selection for certain species?

Line 110: You kept your microbial community in the flocculation section for 7 days. Did you observe the community within this time? Did you ventilated the system or did it become anoxic? Are the natural conditions of the community unaffected by this procedure so that you can compare the aggregation processes in your system with natural aggregation processes? Can you please discuss this in the ms.

Line132-135: This sentence is difficult to understand. Can you write something like this: The threshold line defined overlapping pixels as biological or MP pixels by dividing equally all pixels outside of the 99% confidence limits (Fig. 1).

Line 135: Please define t and p in the equation.

Line 145: Do you mean with terminal velocity modeled/calculated velocity?

Line190: explain with other words a and d, what is a volume-based shape factor of the fractal aggregate and volume fractal dimension?

Line194: What is a fractal dimension and a fractal scaling parameter?

Line 193-201: This section is very difficult to understand. Can you explain in more detail and write it in a simpler way?

Line339-340: Can you rewrite the sentence? It is not understandable.

Is there a reason why do you used brackish water in your experiments instead of marine water with PSU around 35? Did you perform the experiment also with fresh water? Do you will gain different results in sinking behavior dependent on the different biota/aggregate formation in the different water regimes that you cannot calculate in your model? The density of the water is easy to adjust in your model but the salinity can have different impact on biota behavior, formation of particles, stickiness and so on. Can you discuss it in your discussion part?

Did you also test in your OMCEC system the sinking behavior of different types of plastic directly? Are these results comparable with your modeled results?

Minor comments:

The resolution of the figures is very blurry. Can you improve the resolution for the final publication?

Figure 7b: in the legend please change HPDE into HDPE.

6. PLOS authors have the option to publish the peer review history of their article (what does this mean?). If published, this will include your full peer review and any attached files.

Reviewer #1: Yes: Katrin Wendt-Potthoff

Reviewer #2: Yes: Cathleen Schlundt

---

## [Author Response · Author response to Decision Letter 0]

12 Dec 2019

Review comments and response to Journal Requirements

Comment 1. Our internal editors have looked over your manuscript and determined that it may be within the scope of our Plastics in the Environment Call for Papers. The Collection will encompass a diverse range of research articles to better understand various aspects of the effect of plastics in the environment. Additional information can be found on our announcement page: https://collections.plos.org/s/plastics‐environment. If you would like your manuscript to be considered for this collection, please let us know in your cover letter and we will ensure that your paper is treated as if you were responding to this call. If you would prefer to remove your manuscript from collection consideration, please specify this in the cover letter.

Thank you for your consideration, and we would like to include our paper in this special collection.

Comment 2. In your Methods section, please provide additional location information, including geographic coordinates for the data set if available.

Changed accordingly. We added the coordinate of the sampling site (Section Material preparation, line 77).

Comment 3. In your Methods section, please provide additional information regarding the permits you obtained for the work. Please ensure you have included the full name of the authority that approved the field site access and, if no permits were required, a brief statement explaining why.

Our sampling has not required a permit because we only collected plankton in river water in a public area and we confirm that the field study did not involve endangered or protected species. We added this ethics statement in the lines 236-239.

Review comments and response to Reviewer #1

Comment 1. The manuscript deals with an original and intelligent approach to predict the behavior of different microplastic particles associated with different degrees of biological colonization in natural water. The experimental database is very well explained and carefully evaluated. I have to admit that I did not check the equations, but their construction has been explained clearly and understandable for non‐specialists. It should be stated more clearly in

the text where the conclusions are based on the authors’ own experiments as opposed to theoretical simulation. In the latter case, some assumptions need critical evaluation (see below). To my opinion there is already some discussion of findings in the results section, which should be commented by the handling editor.

We thank the Reviewer for reviewing our paper and her positive opinion. We have considered her suggestions and made some corresponding changes to the manuscript. We address her attention to the point-by-point responses given below.

Comment 2. Abstract: line 3 should probably read “excess buoyancy relative to water”.

Changed accordingly (Abstract line 3).

Comment 3. “MPs” in plural form is sometimes incorrect, think of using “MP” throughout.

We now use “MPs” for nouns and “MP” for adjectives throughout the manuscript (changes were not marked in red in the “Revised Article with Changes Highlighted” file).

Comment 4. Last phrase: statement should be expressed more cautiously, as the application is purely theoretical and has limitations (see later).

We agree with the Reviewer, and we have made modification to emphasize that the model is a preliminary tool to estimate the vertical transport of MP aggregates (Abstract lines 14-15).

Comment 5. Lines 2‐3: statement is misleading, the references point to input from rivers, which integrate many pathways of plastic input, among which household sewage systems do not dominate.

Changed accordingly (lines 2-5). We rephrased that plastic production is for household, retail, and industry demands, and the main reason for plastics reaching the oceans is from the mismanaged plastic waste in general. 

Comment 6. Lines 17‐18: plastic does not feed higher trophic levels (not even the level where it enters the food chain) – better: it can be transferred to higher trophic levels.

Changed accordingly (lines 18-19).

Comment 7. Line 84: “cycle” instead of “cycles” – please also give details on the vessel shape and size, the light intensity and incubator type.

Changed accordingly (lines 85-92). Each of the pre-incubated samples (500ml) was contained in a 600 mL glass beaker and was agitated on the orbital shaker EOM5 at 100 rpm, at 20⁰C, under a 9:15 light:dark cycle with light being generated by a 13 W bulb in the laboratory for two weeks.

Comment 8. Line 228: up to 9.x % biological ballast is not “without biological matter”, this only fits for the virgin plastic particles –please adjust statement.

We agreed with the Reviewer and removed this comment (lines 260-261). 

Comment 9. Line 309: it should be said more clearly that this part of the study is mostly based on literature data, not on experimentation.

We changed the section title to “Modelled terminal velocity of low- and high-density MPs” and changed the word “estimate” to “model” and refer the readers to the equation used. We also emphasized that we applied the geometrical properties of polyurethane MP aggregates to all other polymers (lines 341-346).

Comment 10. Line 336: I do not understand how this number is derived from Eriksen et al.’s study, since they reported particle counts and weights on a square kilometer basis. Moreover, the comparison is critical, as Eriksen et al. recorded particles larger than 300 μm, and the study under review used PUR sawdust sized 90‐300 μm. Please explain more clearly.

We appreciated the Reviewer’s comment. The average MP concentration in the ocean 0.003 g m-3 we used was actually taken from Lebreton et al. (2017)’s paper, where they cited Eriksen et al. (2014) as the reference source for that number. They derived the volume concentration (g m-3) by dividing the concentration per surface area (g km-2) to the water depth that the trawling devices reached. We should have cited Lebreton et al. (2017) as well in the manuscript for using that number. We agreed with the Reviewer #1’s comment about the MP size larger than 300 μm in the source and the size smaller than 300 μm in our analysis. Together with the concern of the Reviewer #2 regarding this section, we removed the section “mass flux of low- and high-density MPs” from our manuscript to avoid any confusion to readers (lines 367-376). 

Eriksen, M., Lebreton, L. C., Carson, H. S., Thiel, M., Moore, C. J., Borerro, J. C., ... & Reisser, J. (2014). Plastic pollution in the world's oceans: more than 5 trillion plastic pieces weighing over 250,000 tons afloat at sea. PloS one, 9(12), e111913.

Lebreton, L. C., Van der Zwet, J., Damsteeg, J. W., Slat, B., Andrady, A., & Reisser, J. (2017). River plastic emissions to the world’s oceans. Nature communications, 8, 15611.

Comment 11. Lines 380‐381: please say more clearly that only PUR has been experimentally tested. It may differ in colonization from other polymers due to several factors, one of which is the fact that it is the only polymer containing nitrogen (see e.g. study by Russell et al. 2011, https://aem.asm.org/content/77/17/6076)

We added the polymer type (polyurethane) to the context to clarify our MP type (line 415) and discuss the difference in the colonization of different polymers in the discussion section (lines 465-473). 

Comment 12. Line 411: please check carefully if the given weight is correct. Reference 69 includes plastic of different size classes, and reference 70 reports plastic particles larger than 1 cm throughout. Moreover, even if the particles sink faster, this does not necessarily mean that fish do not catch them, they might be attracted by their motion.

We appreciated the Reviewer’s comment. We did not take into account the size of MPs while giving to this space in discussion. We adjusted our argument to “Aquatic organisms at different depth of the water column, therefore, may have more time and chance to encounter MPs” and remove these references (line 444-450). 

Comment 13. Line 415 and thereafter: regarding the limitations of the study, it should also be considered that the shape characteristics of the other polymers might be different even after using the same sawing procedure due to their physical properties. This also applies to their wettability, which may greatly influence colonization and transport.

We added in our discussion the limitations due to using one type of polymer to represent many different polymers. MPs with different polymers can be various in size, shape, and cell colonization (lines 465-475).

Comment 14. Line 428: faecal or fecal pellets

Changed accordingly (line 464). 

Review comments and response to Reviewer #2

Comment 1. The manuscript „Sinking of microbial‐associated microplastics in natural waters” by Nguyen et al. is original, interesting, timely, and an important contribution to the current knowledge of marine plastic research. The experimental design is elegant and the results provide new insight into the movement behavior of microplastic within the water column. The combination of lab experiments and model approaches expand the knowledge about the movement of different plastic types in the water column in relation to biofouling. The only criticism I have is the way the results are presented. The authors presume that the readers are experts in the research field of the authors. Important explanatory details about the methods are missing. This makes the ms very difficult to read and to understand. The authors used a lot of different parameters, some of them not properly explained, which can easily confuse the reader. I recommend to add more explanation and rewrite some parts of the ms in a simpler way. The paper will gain more visibility when it is easy to understand. I hope my comments below can help to focus on parts that have room for improvements.

We thank the Reviewer for reviewing our paper and her positive opinion. We have revised the manuscript by adding more explanations and restructuring the section describing the terminal velocity equation. We think our methods section has improved substantially and has become easier to follow than before. We address her attention to the point-by-point responses given below.

Comment 2. You present a lot of calculations in your ms with plenty of parameters. Most of the parameters are explained somewhere in the text. However, it is easier to follow the text with having an overview of all parameters. Can you expand Table 1 and include all parameters you used in your equations? Explain in an additional column in the same table the meaning of the parameters. Something like:

Parameter | Units | Values | definition | equation

A | μm | 10‐200 | projected area = sum of image pixels of an aggregate | 1, 2, 3,…

Thanks, we have now listed all parameters used in the equation and explained their values, sources, and definitions in Table 1. 

Comment 3. Line 66‐67: Please explain in more details in the ms what do you mean with “shape irregularity and fractal structure”.

We took into account shape irregularity as the difference in shapes of MPs and MP aggregates using shape factors bL and cL to calculate the area A and perimeter P from size L in the equations. The fractal structure of the aggregates was expressed through the fractal dimension d that follows the fractal scaling against the size L. We have included a more detailed explanation in lines 206 to 215.

Comment 4. Line 75: You used a plankton net with 20μm mesh size and you said that you sampled bacteria (line81). Most bacteria range between 0.2 and 2μm. Do you wanted to exclude most of the bacteria? Why you didn’t use unfiltered water for your pre‐incubation to obtain the entire biota? And why do you pre‐incubated the water before you bring it in the OMCEC system? Please explain in the ms.

The reason we filtered the river water was to exclude fine mineral in the sample. The presence of minerals can affect the size, density, and settling velocity of the observed aggregates in this study, where we focused only on MPs and biological matter. Similarly, we pre-incubated the biological samples to minimize the interference of minerals in our experiment. Mineral particles may exist in our biological samples because they may attach to the biological aggregates larger than 20 μm. Because mineral size is similar to bacteria, it is challenging to exclude it from the biological samples. Hence, we diluted the biological samples at a dilution factor of 5 and we pre-incubate the samples to increase the microbial population that we have (very little at the beginning) to be enough for the main experiment in the settling column (8 L). Through pre-incubation and dilution, we can therefore eliminate fine minerals from our samples. We added this explanation to our methods section in lines 78-80, and 84-87.

Comment 5. Line 85: You added nitrate and glucose to your experiment, why? Do you want to trigger a phytoplankton bloom? Please explain in the ms. In general, can you give more details in your ms why you manipulated the plankton community before you used them in the OMCEC system? Is your community still a natural community or is there a selection for certain species?

As mention above, the reason we pre-incubated the biological sample is to eliminate fine minerals and increase the microbial population for the experiment in a larger volume. The addition of nutrients was to support microbial growth and shorten the experiment period. We were aware that adding nitrate and glucose might have selected some of the species as compared to the natural community. We added a note to this aspect in lines 92-93.

Comment 6. Line 110: You kept your microbial community in the flocculation section for 7 days. Did you observe the community within this time? Did you ventilated the system or did it become anoxic? Are the natural conditions of the community unaffected by this procedure so that you can compare the aggregation processes in your system with natural aggregation processes? Can you please discuss this in the ms.

The flocculation section of the settling column where the incubation occurred is ventilated because the settling column was loosely capped, and an 8 L of head space was available to allow sufficient gas exchange between atmosphere and the water column. In addition, the suspension was sheared for 30 minutes every 2 hours to aerate the suspension (lines 122-125). We did not perform quantitative measurements of the microbial community. 

The settling column was designed to replicate as much as possible the hydrodynamic conditions of natural aqueous environments like turbulence shear rates, the use of river water (with nutrient addition), and lighting system. However, the experiment is still laboratory-based, and thus the aggregation processes may be different from the natural aggregation processes. On the one hand, we agree that this is a disadvantage of the experiment but, on the other hand, this allows replication of experiments at controlled conditions for comparison with other experiments of MP associating with biological matter in the literature (Long et al., 2015 and Porter et al., 2018) that are also laboratory based. We added a small discussion of the use of the settling column to minimize the disturbance to the flocculation processes in lines 118-120. 

Long, M., Moriceau, B., Gallinari, M., Lambert, C., Huvet, A., Raffray, J., & Soudant, P. (2015). Interactions between microplastics and phytoplankton aggregates: Impact on their respective fates. Marine Chemistry, 175, 39-46.

Porter, A., Lyons, B. P., Galloway, T. S., & Lewis, C. (2018). Role of marine snows in microplastic fate and bioavailability. Environmental science & technology, 52(12), 7111-7119.

Comment 7. Line132‐135: This sentence is difficult to understand. Can you write something like this: The threshold line defined overlapping pixels as biological or MP pixels by dividing equally all pixels outside of the 99% confidence limits (Fig. 1).

We rephrased this sentence in line 149-152. We did not use the suggested sentence because the threshold line did not equally divide the overlapping pixels, instead, it is the bisector of the two 99% confidence limits.

Comment 8. Line 135: Please define t and p in the equation.

Gt + Bt is the threshold value calculated using the threshold equation and is compared with Gp + Bp, which is the green and red intensities of a pixel, to determine the material of that pixel. The use of Gt + Bt may be confusing, and thus we have changed it to I instead of Gt + Bt (lines 151-153). 

Comment 9. Line 145: Do you mean with terminal velocity modeled/calculated velocity?

Terminal velocity is the velocity of an object when all the forces applied on it reach a balance ΣF = 0. Terminal velocity can be settling velocity (downward) and rising velocity (upward). We used the term “settling velocity” in the experiment because we know the direction of the velocity is downward, while the term “terminal velocity” was used in the model for a general situation because MPs can either settle, rise, or float. We rephrased the section title and the first sentence of the section to clarify the definition of terminal velocity (line 163-165).

Comment 10. Line190: explain with other words a and d, what is a volume‐based shape factor of the fractal aggregate and volume fractal dimension?

We simplified “a” as volume-based shape factor of the aggregate, and “d” as fractal dimension of the aggregate. We used the set of shape factors, including “a” – volume-based, “b” – area-based, and “c” – perimeter-based to show how different the aggregate external shape as compared to a cube and a square. The fractal dimension d, on the other hand, represents the aggregate internal structure and compactness. The implementations are in line 213-215 and 217-219.

Comment 11. Line194: What is a fractal dimension and a fractal scaling parameter?

Fractal dimension d is a parameter ranging from 1 to 3, representing the internal structure of the aggregate with a higher value of d representing a more compact structure (Maggi and Winterwerp, 2004). The fractal dimension can be scaled with aggregate size L using the scaling law suggested in Maggi (2007), where γ is the fractal scaling parameter that shows the rate of change in d over the aggregate dimensionless size L/Lp. We added these explanations in lines 213-215, 218-219, and 222-223.

Maggi, F., & Winterwerp, J. C. (2004). Method for computing the three-dimensional capacity dimension from two-dimensional projections of fractal aggregates. Physical Review E, 69(1), 011405.

Maggi, F. (2007). Variable fractal dimension: A major control for floc structure and flocculation kinematics of suspended cohesive sediment. Journal of Geophysical Research: Oceans, 112(C7).

Comment 12. Line 193‐201: This section is very difficult to understand. Can you explain in more detail and write it in a simpler way?

We rephrased these sentences in line 223-225. The application of the Heaviside function is to avoid the fractal dimension of virgin MP particles to change with their size, because virgin MP particles are compact and should have d = 3.

Comment 13. Line339‐340: Can you rewrite the sentence? It is not understandable.

This section was deleted as it was concerned by both Reviewers (367-376).

Comment 14. Is there a reason why do you used brackish water in your experiments instead of marine water with PSU around 35? Did you perform the experiment also with fresh water? Do you will gain different results in sinking behavior dependent on the different biota/aggregate formation in the different water regimes that you cannot calculate in your model? The density of the water is easy to adjust in your model but the salinity can have different impact on biota behavior, formation of particles, stickiness and so on. Can you discuss it in your discussion part?

The sampling location followed from the sampling campaign organized by the New South Wales Office of Environment and Heritage. We had similar experiments with freshwater before but for microbial-associated mineral aggregates, not microplastics (Nguyen et al., 2017; Tang and Maggi, 2016). Our previous results with microbial-associated mineral aggregates showed that different types of microorganisms (e.g., with and without secreting exopolymer products) at different nutrient concentrations, shear rates, mineral concentrations would form aggregates with different size, fractal dimension, and settling velocity. We added this point in the discussion (lines 473-481) and suggested more experiments on different types of microorganisms at various environmental conditions to give a better understanding of this complex mechanism.

Nguyen, T. H., Tang, F. H., & Maggi, F. (2017). Optical measurement of cell colonization patterns on individual suspended sediment aggregates. Journal of Geophysical Research: Earth Surface, 122(10), 1794-1807.

Tang, F. H., & Maggi, F. (2016). A mesocosm experiment of suspended particulate matter dynamics in nutrient-and biomass-affected waters. Water research, 89, 76-86.

Comment 15. Did you also test in your OMCEC system the sinking behavior of different types of plastic directly? Are these results comparable with your modeled results?

We did not test different types of polymers with our OMCEC system, but we expected some differences between them because MPs of different polymers will have different sizes, shapes, formulas, linkages, and additives, even with the same generation methods. Therefore, our equation is suggested to be only a preliminary tool and should be validated with more experimental data for different types of polymers and microorganisms. This point was added in the last sentence of the Abstract and in the discussion section (lines 465-473).

Comment 16. The resolution of the figures is very blurry. Can you improve the resolution for the final publication?

The low resolution is due to the journal compiling process. We submitted high-resolution figures for the revision.

Comment 17. Figure 7b: in the legend please change HPDE into HDPE.

The figure was removed together with the section “Mass flux of low- and high-density MPs.”

---

## [Editor Report · Decision Letter 1]

10 Jan 2020

Sinking of microbial-associated microplastics in natural waters

PONE-D-19-30091R1

Dear Dr. Nguyen,

We are pleased to inform you that your manuscript has been judged scientifically suitable for publication and will be formally accepted for publication once it complies with all outstanding technical requirements.

With kind regards,

Matthäus Bäbler, Ph.D.

Academic Editor

PLOS ONE
---

## [Editor Report · Acceptance letter]

15 Jan 2020

PONE-D-19-30091R1 

Sinking of microbial-associated microplastics in natural waters 

Dear Dr. Nguyen:

I am pleased to inform you that your manuscript has been deemed suitable for publication in PLOS ONE. Congratulations! Your manuscript is now with our production department. 

With kind regards,

on behalf of

Dr. Matthäus Bäbler 

Academic Editor

PLOS ONE